# The Relationship between Soil Respiration and Plant Community Functional Traits in Ebinur Lake Basin

**Huiyi Sheng [1,2], Jinlong Wang [1,2], Xuemin He [1,2] and Guanghui Lv [1,2,*]**

[1] College of Ecology and Environment, Xinjiang University, Urumqi 830046, China; shy@stu.xju.edu.cn (H.S.); wangjl45@stu.xju.edu.cn (J.W.); hxm@xju.edu.cn (X.H.)

[2] Key Laboratory of Oasis Ecology of Education Ministry, Xinjiang University, Urumqi 830046, China

[*] Correspondence: ler@xju.edu.cn; Tel.: +86-13999216886

**Abstract:** Soil respiration (Rs) plays an important role in the carbon cycle of terrestrial ecosystems. Understanding the impacts of plant community functional traits on Rs is a key prerequisite for accurate prediction of the future carbon balance of terrestrial ecosystems under climate change. This study examined the relationship of Rs with plants in the Ebinur Lake Basin in the arid desert region. Traditional statistical methods and geostatistical methods were used to study the spatial variation characteristics of Rs and to analyze the effects of plant community functional traits and environmental factors on the spatial heterogeneity of Rs. The variation in Rs in the arid desert area of the Ebinur Lake Basin showed a strong spatial dependence ($C_0/(C + C_0) = 0.11$) and a medium variation ($\alpha = 25.50$, CV = 86.21%). Principal component analysis revealed that morphological traits of plants and soil water content had great contributions to PC1, soil nutrient had great contributions to PC2, and physiological traits of plants and soil temperature had large contributions to PC3. Multiple regression analysis showed that PC1, PC2, and PC3 can explain 83% of the spatial heterogeneity of Rs ($F = 157.41$, $p < 0.01$). In conclusion, maximum height, leaf width, leaf dry matter content, leaf thickness, and soil water content were the principal driving elements of soil respiration heterogeneity.

**Keywords:** arid desert area; Ebinur Lake Basin; soil respiration; plant functional traits

## 1. Introduction

Soil respiration (Rs) is the main export pathway of organic carbon from soil carbon pools and a major source of atmospheric carbon dioxide [1]. Rs contributes approximately 10% of the total atmospheric carbon dioxide [2], which is the second largest flux of carbon cycle in terrestrial ecosystems [3]; the tiny change in Rs has a significant impact on the global carbon cycle; and Rs occupies an important position in global carbon budget.

Studies have shown that Rs is regulated by many biological and abiotic factors [4]. These biological factors include community composition [5], vegetation types [6,7], stand structure [8], underground/aboveground biomass of annual plant functional groups [9], and litterfall [10,11] from vegetation, where litter quantity and quality supplied to soil affect soil microclimate and influence Rs [7], the patterns of photosynthate partitioning amongst belowground components, the ecosystem level effects of individual plant traits, and the importance of material cycle and species invasions or extinctions in ecosystem [12]. Additionally, abiotic factors include soil temperature (ST) and soil water content (SWC) [13–17]. Soil organic content (SOC) at a certain depth is closely related to $CO_2$ annual flux [18–23]. Studies have shown that the binary linear regression equation controlled by SOC and total nitrogen (TN) can explain 92.5% of the spatial heterogeneity of Rs [24]. In arid desert regions, however, SWC has the highest explanatory power for the spatial heterogeneity of Rs [25,26].

Plant functional traits are key variables for measuring plant community response to environmental change [27], and they can be effective predictors of changes in community structure and ecosystem processes [28,29]. Previous studies on plant functional

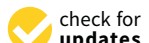

traits revealed that plants changed their morphology and life history traits to environment changes [30], such as leaf carbon content (LCC), leaf nitrogen content (LNC), specific leaf area (SLA) [31], leaf length (LL), leaf width (LW), leaf thickness (LT), and seed size. These occur in plants as independent or synergistic ecosystem responses to environmental changes [32]. Moreover, species with similar ecological niches have similar competitive abilities, and the complementary or cross-cutting nature of plant functional traits in response to environmental changes reflects the fitness of individuals within a community, leading to coexistence between species and exclusion between non-species. As the largest and most environmentally sensitive organ, the leaf is the basic unit of material, energy, and water vapor exchange in the ecosystem and is often the subject of study [33]. In summary, plant traits play an important role in physiological processes such as nutrients utilization, water absorption and transport, growth, reproduction, and self-defense. However, to date, few comprehensive studies on the relationship between plant functional traits and Rs have been conducted.

Land degradation, desertification, salinization, reduced biodiversity, and altered community stability in desert ecosystems are affected by changes in the extreme climate and precipitation landscape worldwide [34]. The Ebinur Lake Basin is at the center of mainland Asia, with a dry climate and strong water evaporation. Its species composition and distribution are extremely sensitive to external disturbance, but it has special and precious biological resources in the harsh and complex growth conditions and has unique conditions in terms of wind and sand fixation, dust reduction and sand removal, water conservation, and maintenance of oasis diversity and stability, which affect the living environment of the surrounding areas. Studying the spatial heterogeneity of Rs and its influencing factors under the background of global change is of great significance to estimate and predict the changes of Rs in Ebinur Lake Basin and to quantitatively study the underground carbon flux and carbon distribution pattern in an arid desert ecosystem.

This study examined the functional traits of plant communities and environmental factors to analyze the influencing factors and spatial variation characteristics of Rs, which has important theoretical value for revealing the carbon cycle mechanism of arid desert ecosystem and can provide a theoretical basis for the relationship between community characteristics and ecosystem function.

## 2. Materials and Methods

### 2.1. The Study Area

The study area is in the Ebinur Lake Basin National Nature Reserve, with a geographical position of 82°36′–83°50′ E, 44°30′–45°09′ N, in the hinterland of the Eurasian continent, with a total basin area of 2670.85 km$^2$ [35]. It belongs to the desert ecosystem of a temperate arid zone. The climate is extremely dry, rainless, and windy. Its average annual rainfall is 105 mm, the evapotranspiration is 1315 mm, the average temperature is 6 °C–8 °C, and the daily and annual differences in temperature are large [36].

Due to the unique climatic conditions, extreme abiotic factors, hydrological processes, and geographical location, the vegetation types in this area are relatively single, and the community structure and ecosystem structure are fragile. The main dominant species include *Populus euphratica*, *Haloxylon ammodendron*, *Reaumuria soongorica*, *Halimodendron halodendron*, *Nitraria schoberi*, and *Suaeda* [37]. Soil salinization is intense and alkalization is obvious in this basin [38]. *Phragmites australis*, *H. ammodendron*, *H. halodendron*, *N. schoberi*, *P. euphratica*, and *Apocynum venetum*, among others, have shown strong tolerance to salt and alkali drought [39], which laid an important foundation for the construction of unique communities in the region.

### 2.2. Sample Square Layout

This study examined a typical desert ecosystem in the Ebinur Lake Basin, where a large sample plot of 1 ha (100 m × 100 m) was set up 150 m away from the bank of the Achiksu River to the north of the East Bridge Management and Protection Station (Figure 1). The

selected large sample plots were divided into 100 samples of 10 m × 10 m and 400 samples of 5 m × 5 m continuous plot units.

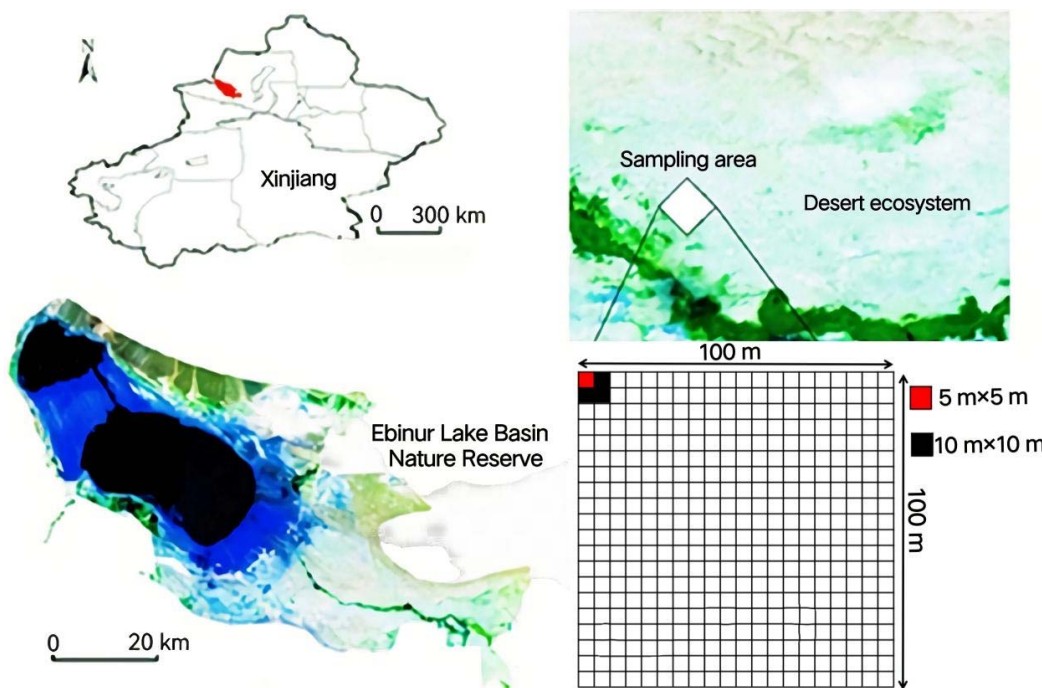

**Figure 1.** Layout of the sample plot.

*2.3. Sample Collection and Determination*

(1)    Determination of the soil respiration

This study used a 10 m × 10 m scale to monitor Rs using the automatic soil $CO_2$ flux system (LI-8100, LI-COR, Lincoln, NE, USA). The center point of each sample square was the monitoring point of Rs. Before obtaining measurements, each sample square was numbered, and the relative spatial coordinate value of each sample point was recorded. To reduce the impact of PVC placement on Rs, two PVC rings were placed at each sampling point 1 day before sampling. To reduce the time required for monitoring, two LI-8100 were used simultaneously. Each measurement was repeated three times. To avoid the direct impact of precipitation and temperature changes, we conducted Rs monitoring from 10:00 to 12:00 in fine weather when the temperature was within 35 °C ± 1 °C. To eliminate the interference of ground plants on Rs, large branches and surface vegetation were removed from the measuring system ring before the monitoring day. Each sample point was monitored three times, with each Rs measurement at 120 s and 90 s for the total and effective measurements, respectively. During the measurement process, to ensure a leak-free air chamber, the air chamber was carefully and tightly fastened to the PVC ring; then, three Rs measurements were taken and the average value obtained was recorded.

(2)    Determination of environmental factors

This study monitored soil temperature using soil temperature and moisture meter in the scale of 10 m × 10 m (5TE, Decagon, Pullman, WA, USA). The soil samples of 0–10 cm was mixed in each quadrat, and the soil samples were collected in an aluminum box and brought back to the laboratory for determination of physical and chemical properties. Soil moisture content (SWC) was determined using the drying method; SOC and TN were determined using the potassium dichromate volumetric method and the Kjeldahl method, respectively.

(3)   Investigation, determination, and calculation of plant functional traits

The information of the species composition features based on the field survey and statistics are shown in Table 1. The ecological significances of selecting plant functional traits indicators are showed in Table 2. In the 5 m × 5 m sample units, the species names, number, maximum height (MH), LL, LW, and LT of trees and shrubs were recorded. Herbaceous plants were observed in the 5 m × 5 m sample units with three 1 m × 1 m sample units using the diagonal method. Using a vernier caliper, the LL, LW, LT, and leaf area of five healthy leaves selected from all species in each sample plot were obtained [40]. To approximate the cylindrical-shaped leaf of *Haloxylon ammodendron*, a vernier caliper was used to measure the leaf diameter and LL; then, the total area of the blade was computed using the formula for calculating the surface area of a cylinder [41]. The fresh weights of leaves were determined using an analytical balance to weigh the freshly picked plant leaves stored in an envelope bag of known weight. Dried weights were measured after oven-drying the leaves in a pre-heated oven at a temperature of 100 °C–105 °C. The bagged leaves were placed in the oven for 10 min and dried at a constant weight at a lowered oven temperature of 70 °C–80 °C. The leaves were cooled off inside a desiccator with a constant temperature and then weighed with a 1/10,000 analytical balance. The SLA and LDMC were calculated using the following formulae:

$$\text{SLA} = \frac{\text{Leaf area}}{\text{Leaf dry weight}} \tag{1}$$

$$\text{LDMC} = (\text{Leaf dry weight})/(\text{Leaf saturated fresh weight}) \times 100\% \tag{2}$$

The LCC, LNC, and LPC were also measured from leaves that were pulverized on a grinder (MM400, Retsch, Ottohahn, Germany). The LCC, LNC, and LPC were determined using the potassium dichromate volumetric method, the Kjeldahl method, and molybdenum antimony resistance colorimetry [42], respectively.

**Table 1.** List of plants in the sample plots at the Ebinur Lake Basin.

| Scientific Name of Plant | Family | Genus | Biotype |
|---|---|---|---|
| *Populus euphratica* Oliv. 1807 | Salicaceae | Populus | arbor |
| *Haloxylon ammodendron* (C. A. Mey.) Bunge. 1851 | Chenopodiaceae | Haloxylon | dungarunga |
| *Halimodendron halodendron* (Pall.) Voss. 1896 | Leguminosae | Halimodendron | bush |
| *Nitraria tangutorum* Bobr. 1946 | Zygophyllaceae | Nitraria | bush |
| *Lycium ruthenicum* Murray. 1780 | Solanaceae | Lycium | bush |
| *Apocynum venetum* Linn. 1753 | Apocynaceae | Apocynum | fruticuli |
| *Alhagi sparsifolia* Shap. 1933 | Leguminosae | Alhagi | fruticuli |
| *Reaumuria soongorica* (Pall.) Maxim. 1889 | Tamaricaceae | Reaumuria | fruticuli |
| *Phragmites australis* (Cav.) Trin. 1841 | Gramineae | Phragmites | herbaceous |
| *Suaeda microphylla* (C. A. Mey.) Pall. 1803 | Chenopodiaceae | Suaeda | herbaceous |
| *Achnatherum splendens* (Trin.) Nevski. 1937 | Gramineae | Achnatherum | herbaceous |
| *Salsola collina* Pall. 1803 | Chenopodiaceae | Salsola | herbaceous |
| *Mulgedium tataricum* (Linn.) DC. 1838 | Compositae | Mulgedium | herbaceous |
| *Glycyrrhiza uralensis* Fisch. 1825 | Leguminosae | Glycyrrhiza | herbaceous |
| *Sonchus oleraceus* Linn. 1753 | Compositae | Sonchus | herbaceous |

**Table 2.** Nine plant functional trait indicators, unit of measure, and their ecological significance.

| Plant Functional Traits | Unit | Ecological Significance |
|---|---|---|
| Maximum height, MH | cm | Reflects the light competitiveness of plants, such as growth form, position of the species in the vertical light, gradient of the vegetation, competitive vigour, reproductive size, whole-plant fecundity, potential lifespan [43] |
| Leaf length, LL | mm | LL is connect with the area of a leaf, which can be linked to allometric factors (plant size, twig size) and ecological strategy with respect to environmental nutrient stress [43] |
| Leaf width, LW | mm | LW is connect with the area of a leaf, which can be linked to allometric factors (plant size, twig size) and ecological strategy with respect to environmental nutrient stress [43] |
| Leaf thickness, LT | cm | Reflects the ability to resist physical damage [43] |
| Leaf dry matter content, LDMC | $g \cdot g^{-1}$ | Reflects the ability of plants to obtain resources and to resist physical damage [43,44] |
| Specific leaf area, SLA | $cm^2 \cdot g^{-1}$ | Reflects the ability of plants to obtain resources, correlated with photosynthetic intensity [43,45] |
| Leaf carbon content, LCC | $mg \cdot g^{-1}$ | Reflects the photosynthetic intensity [43] |
| Leaf nitrogen content, LNC | $mg \cdot g^{-1}$ | Reflects carbon sequestration and soil fertility maintenance capacity [46] |
| Leaf phosphorus content, LPC | $mg \cdot g^{-1}$ | Reflects the osmotic regulation ability, stress resistance, and adaptability of plants [47] |

*2.4. Data Processing and Analysis*

(1) The community weight means (CWM) is used to represent the average value of community functional traits of all species in the community. CWM is weighted as the sum of leaf functional traits and relative abundance of all species in each quadrat [48], as follows:

$$T_c = \sum_1^k P_j T_j \tag{3}$$

where $T_c$ is the feature value of leaves' functional traits in each plot, $P_j$ is the weight of species $j$ in each plot, $T_j$ is the feature value of leaves' functional traits of species $j$ in each plot, and $k$ is the number of species in each plot [48].

(2) According to the analysis of classical statistical methods, the percentage of the ratio of standard deviation and average value of the sample is the coefficient of variation (CV) of the sample. The research variables were determined using descriptive statistics, and their coefficient of variation (CV) was calculated with the following formula:

$$CV = \frac{SD}{mean} \times 100\% \tag{4}$$

where CV is the coefficient of variation; $CV \leq 0.1$ indicates that the research variable is weak variation, $0.1 < CV < 1$ indicates medium variation, and $CV \geq 1$ indicates strong variation [49]; SD is the standard deviation; and mean is the mean value.

(3) The semi-variance function was used to describe the spatial variation and correlation degree of local regional variables. The calculation formula of variation function $\gamma(h)$ in geostatistics is as follows:

$$\gamma(h) = \frac{1}{2N(h)} \sum_{i=1}^{N(h)} [Z(x_i) - Z(x_i + h)]^2 \tag{5}$$

where *h* is the separation distance; $N(h)$ is the numerical logarithm of spacing *h*; and $Z(x_i)$ and $Z(x_i + h)$ are the values of the research variables at points $x_i$ and $x_i + h$, respectively.

The range of the semi-variance function model refers to the distance between the variation function and sill, and the range reflects the scale or spatial autocorrelation scale of the spatial heterogeneity [50]. The greater the range value, the weaker heterogeneity; the smaller the range value, the stronger heterogeneity. Moreover, the parameter variation is stronger in a local small range, and the overall parameter distribution is more complex [51].

The parameters of the variation function include nugget ($C_0$), range ($\alpha$), and sill ($C_0 + C$). The nugget denotes the variation in the research object caused by a smaller sampling scale and experimental error; range represents the scope of spatial correlation of research objects. With an increase in spacing h, when the variation function reaches a relatively stable constant from the initial nugget value, this constant is called the sill. The ratios of nugget ($C_0$) to sill ($C + C_0$) were called the nugget effect, which can reflect the proportion of spatial variation caused by random factors in the overall variation. When the nugget value is lower than 0.25, it indicates that the research variable has a strong spatial dependence; when the nugget value accounts for 25% to 75%, it indicates that the spatial variability is in moderate spatial dependence. When the nugget value is greater than 0.75, it indicates that the research variable has a weak spatial dependence [52].

Data collation, statistical analysis, and graphic drawing were performed in Microsoft Excel 2013, GS$^+$ 9.0, SPSS 19.0, and Origin 9.0. After the logarithmic transformation of Rs and the data of each functional trait, the data were descriptively analyzed using SPSS 19.0. The semi-variance function of the research variables was analyzed by GS$^+$ 9.0, and different models were fitted. The determination coefficients and residual sum of squares of different models were compared to obtain the optimal semi-variance function model of each research variables. The spatial distribution maps of each research variables were generated in Origin 9.0. The data matrix was established by functional traits and environmental factors for principal component analysis by SPSS 19.0, and Pearson correlation analysis was used by Origin 9.0 to further analyze the relationship between each research variables. Finally, multiple regression was used by SPSS 19.0 to establish the Rs prediction model.

## 3. Results

### 3.1. Statistical Description of Soil Respiratory and Community Functional Traits

The CV can reflect the degree of variation between the overall and internal data. Table 3 shows that the Rs fluctuated in the range of 0.04–1.11 $\mu mol \cdot m^{-2} \cdot s^{-1}$, with an average value of 0.29 $\mu mol \cdot m^{-2} \cdot s^{-1}$. From the CV value, it can be seen that Rs showed medium variation, CV = 86.21%. MH, LL, LW, LT, SLA, SWC, TN, and SOC showed medium variation in the range of 15.15–32.50%. LDMC, LCC, LNC, LPC and ST showed weak variation in the range of 7.44–9.82%.

Judging from the nugget value, except LCC, the nugget values of other parameters were all below 0.25; this indicated that the spatial variability had a strong spatial correlation. The nugget value of LCC was greater than 0.75, indicating that the autocorrelation of LCC was not strong.

From the perspective of the fitting model, the best fitting models of LT and LPC were the Spherical model; the best fitting model of SLA was the Gaussian model; and the best fitting models of Rs, MH, LL, LW, LDMC, LNC, ST, SWC, and SOC were the Exponential model. The semi-variance function model fitting of each variables showed that the decision coefficients of Rs, MH, LW, LT, LDMC, SLA, LNC, ST, SWC, and SOC were between 0.60 and 0.96, indicating that the sampling spacing and the fitting model were in line with the statistical requirements and that the selected model and the theoretical model had high fitting accuracy and can reflect the spatial structure characteristics of the selected parameters well. LCC can be fitted using the Linear model, and TN can be fitted by the Exponential model. However, the coefficient of determination was low, indicating that the sampling points showed strong randomness.

From the viewpoint of the range, LCC had the weakest heterogeneity ($\alpha$ = 52.97. CV < 10%), followed by LNC, ST, LDMC, and LPC (8.1 < $\alpha$ < 25.5, CV < 10%) and by SOC, LW, Rs, and MH (21.6 < $\alpha$ < 30.00, CV > 10%); TN had the strongest heterogeneity ($\alpha$ = 8.10, CV > 10%), followed by SWC, LT, LL, and SLA (15.0 < $\alpha$ < 19.75, CV > 10%).

TN had the strongest heterogeneity ($\alpha$ = 8.10, CV > 10%), followed by SWC, LT, LL, and SLA (15.0 < $\alpha$ < 19.75, CV > 10%); by MH, Rs, LW, and SOC (21.6 < $\alpha$ < 30.00, CV > 10%); and by LPC, LDMC, ST, and LNC (8.1 < $\alpha$ < 25.5, CV < 10%). LCC had the weakest heterogeneity ($\alpha$ = 52.97. CV < 10%).

**Table 3.** Descriptive statistics and the parameters of theoretical models for soil respiration, plant functional traits, and environmental factors.

| Research Variables | Means ± Standard Deviation | Coefficient of Variation (%) | Variogram Model Type | Range (m) | Nugget Values $C_0/(C + C_0)$ | $R^2$ |
|---|---|---|---|---|---|---|
| Soil respiration ($\mu$mol m$^{-2}$·s$^{-1}$) | 0.29 ± 0.25 | 86.21 | Exponential | 25.50 | 0.11 | 0.60 |
| Maximum strain height (cm) | 135.74 ± 21.12 | 15.56 | Exponential | 21.60 | 0.06 | 0.61 |
| Leaf length (mm) | 148.11 ± 29.30 | 19.78 | Exponential | 19.50 | 0.07 | 0.37 |
| Leaf width (mm) | 15.31 ± 3.32 | 21.69 | Exponential | 27.60 | 0.16 | 0.84 |
| Leaf thickness (cm) | 0.33 ± 0.05 | 15.15 | Spherical | 17.10 | 0.03 | 0.89 |
| Leaf dry matter content (g·g$^{-1}$) | 0.39 ± 0.28 | 8.21 | Exponential | 18.30 | 0.07 | 0.71 |
| Specific leaf area (cm$^2$·g$^{-1}$) | 70.60 ± 14.23 | 20.16 | Gaussian | 19.75 | 0.11 | 0.96 |
| Leaf carbon content (mg·g$^{-1}$) | 407.09 ± 30.27 | 7.44 | Linear | 52.97 | 1.00 | 0.03 |
| Leaf nitrogen content (mg·g$^{-1}$) | 25.20 ± 2.46 | 9.76 | Exponential | 30.90 | 0.15 | 0.86 |
| Leaf phosphorus content (mg·g$^{-1}$) | 2.24 ± 0.22 | 9.82 | Spherical | 16.80 | 0.04 | 0.37 |
| Soil temperature (°C) | 27.36 ± 2.05 | 7.49 | Exponential | 25.50 | 0.16 | 0.65 |
| Soil water content (g·g$^{-1}$) | 13.12 ± 2.33 | 17.76 | Exponential | 15.00 | 0.08 | 0.64 |
| Total nitrogen (g·kg$^{-1}$) | 2.05 ± 0.39 | 19.02 | Exponential | 8.10 | 0.09 | 0.01 |
| Soil organic content (g·kg$^{-1}$) | 9.57 ± 3.11 | 32.50 | Exponential | 30.00 | 0.11 | 0.91 |

*3.2. Spatial Distribution Rules of Soil Respiration, Community Functional Traits, and Environmental Factors*

Figure 2 shows that Rs ($\alpha$ = 25.50, CV > 10%) fluctuated in the range of 0.04–1.11 $\mu$mol·m$^{-2}$·s$^{-1}$ based on data gathered from the Ebinur Lake Basin Nature Reserve. Rs presented "S" type irregular patchy distributions generally. The region with a higher content was in the middle of the north–south direction, and the east and west sides of the region had lower contents. The spatial distribution laws of Rs were similar to those of MH and LW (21.6 < $\alpha$ < 27.6, CV > 10%); however, the maxima and minima of LDMC had exactly opposite laws to Rs. In addition, the Rs was similar to the spatial distribution patterns of LL, LT, SLA, LCC, LNC, LPC, SWC, ST, and TN, and it was similar to them in overlapping patches but was differences in some areas (Figure 2). TN, LT, SOC, and LPC showed irregular patchy distribution, showing high degrees of fragmentation; SWC, LL, SLA, LDMC, ST, LNC, and LCC displayed patchy and blocky distributions, showing uniformity, relatively.

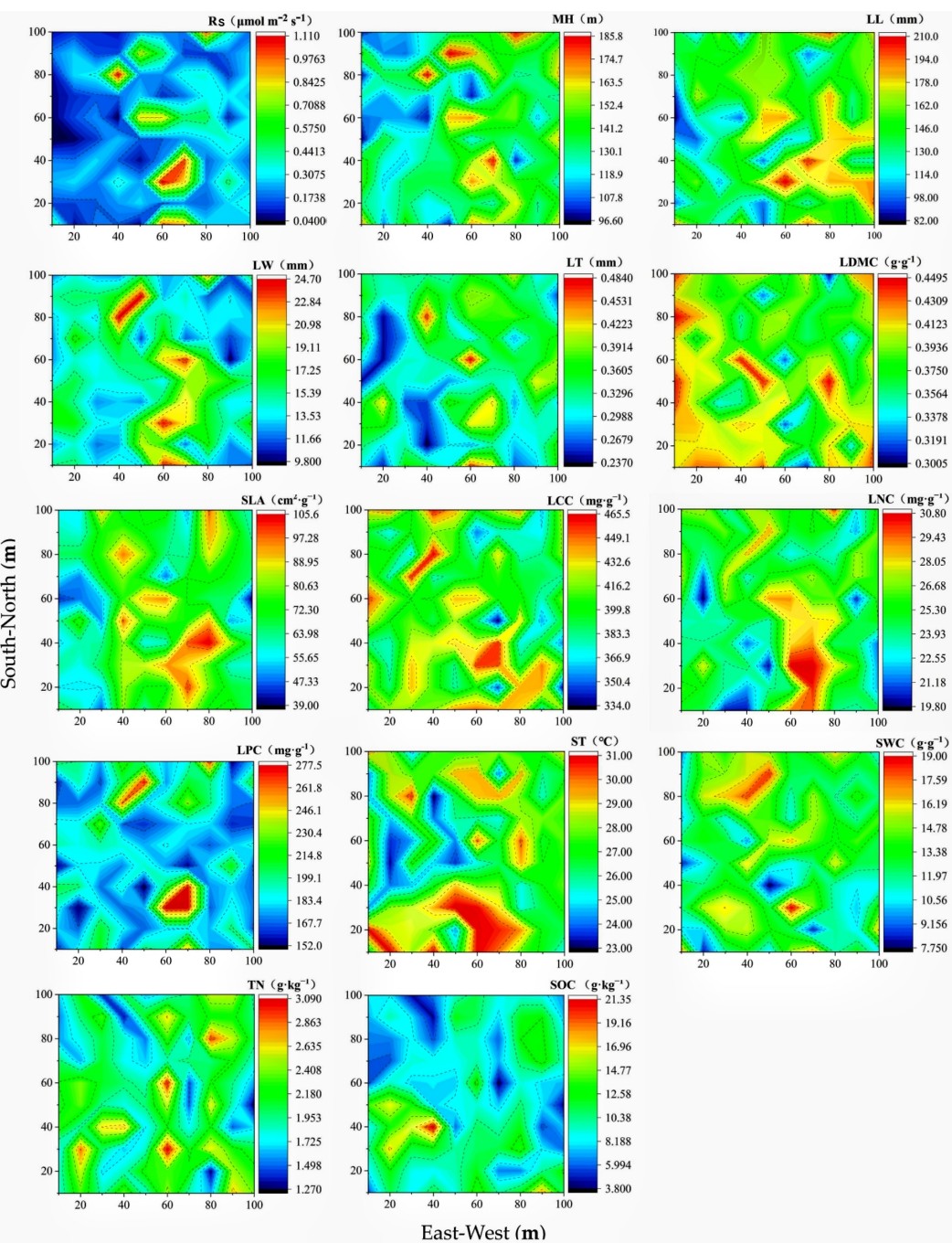

**Figure 2.** Spatial distribution of soil respiration (Rs), community functional traits, and environmental factors (Legend: MH—Maximum height, LL—Leaf length, LW—Leaf width, LT—Leaf thickness, LDMC—Leaf dry matter content, SLA—Specific leaf area, LCC—Leaf carbon content, LNC—Leaf nitrogen content, LPC—Leaf phosphorus content, ST—Soil temperature, SWC—Soil water content, TN—Total nitrogen, and SOC—Soil organic content).

### 3.3. Relationship between Soil Respiration, Community Functional Traits, and Environmental Factors

Principal component analysis was carried out on Rs, environmental factors, and community functional traits. The results showed that the cumulative contribution of PC1, PC2, and PC3 to all the research variables reached 55.48% (Table 4), indicating that PC1, PC2, and PC3 can represent all of the research variables well.

PC1 can explain 34.0% of the total variance in the research variables, with Rs, MH, LW, LDMC, LL, LT, SLA, and SWC contributing a lot to PC1 and with PC1 having a negative

correlation with LDMC and a positive correlation with other research variables. PC2 can explain 12.2% of the total variance in the research variables, with TN and SOC contributing a lot to PC2 and PC2 having a positive correlation with TN and SOC. PC3 can explain 9.2% of the total variance in the research variables, with LCC, LPC, LNC, and ST contributing a lot to PC3 and PC3 having a positive correlation with LCC, LPC, LNC, and ST.

**Table 4.** Total variance explained in soil respiration, plant functional traits, and environmental factors.

| Component | Initial Eigen Values | | |
|---|---|---|---|
| | Total | Variance % | Cumulative % |
| 1 | 4.76 | 34.02 | 34.02 |
| 2 | 1.71 | 12.23 | 46.25 |
| 3 | 1.29 | 9.23 | 55.48 |
| 4 | 1.02 | 7.31 | 62.79 |
| 5 | 0.96 | 6.84 | 69.63 |
| 6 | 0.79 | 5.62 | 75.25 |
| 7 | 0.69 | 4.92 | 80.17 |
| 8 | 0.57 | 4.10 | 84.27 |
| 9 | 0.54 | 3.83 | 88.10 |
| 10 | 0.47 | 3.36 | 91.46 |
| 11 | 0.41 | 2.95 | 94.41 |
| 12 | 0.37 | 2.63 | 97.04 |
| 13 | 0.30 | 2.15 | 99.19 |
| 14 | 0.11 | 0.81 | 100.00 |

A Pearson correlation analysis was performed on Rs, plant functional traits, environmental factors, and three principal components at the community level, as shown in Figure 3. Rs was significantly positively correlated with PC1 and PC3 ($p < 0.01$) but had no significant correlation with PC2 ($p > 0.05$). PC1 was significantly positively correlated with MH, LL, LW, LT, SLA, LNC, LPC, and SWC ($p < 0.01$) and significantly negatively correlated with LDMC ($p < 0.01$), Among them, the correlation coefficients between PC1 and MH, LW, LDMC, and SWC were higher ($R^2 > 0.68$). PC2 was significantly positively correlated with TN and SOC ($p < 0.01$, $R^2 > 0.90$). PC3 was significantly positively correlated with MH, LW, LCC, LNC, LPC, and ST ($p < 0.01$) and significantly positively correlated with LL, LT, LDMC, SLA, and SWC ($p < 0.05$). Among them, the correlation coefficients between PC3 and LCC, LNC, LPC, and ST were relatively higher ($R^2 > 0.51$).

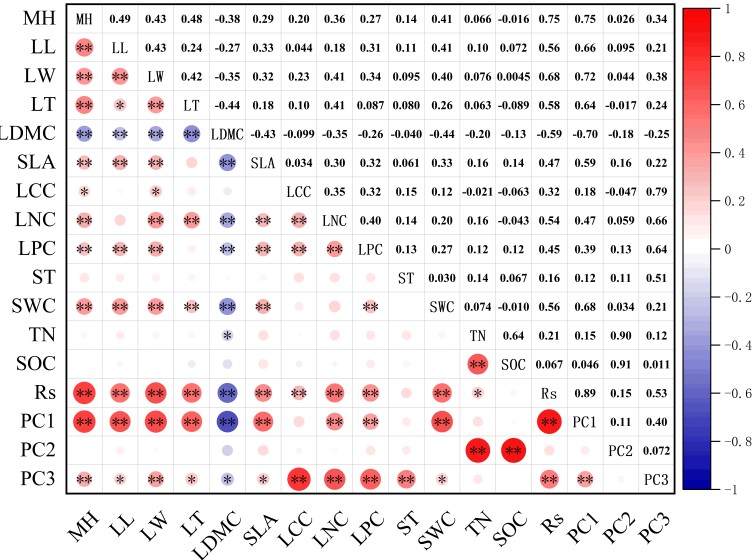

**Figure 3.** Correlation analysis between plant functional traits, environmental factors, and principal components (Legend: $p \leq 0.05$ *, $p \leq 0.01$ **).

*3.4. Multiple Regression Analysis of Soil Respiration (Rs) and Principal Components*

Multiple regression analysis was performed to further analyze the relationship and to establish a regression model between Rs and principal components based on data gathered at the Ebinur Lake Basin, an arid desert area. In Formula (6), the determination coefficient of Rs reached 83% ($F = 157.41$, $p < 0.01$) when the independent variables were PC1, PC2, and PC3.

$$Rs = 0.57PCA1 + 0.04PCA2 + 0.18PCA3 + 9.33 \tag{6}$$

## 4. Discussion

*4.1. Spatial Distribution Characteristics of Soil Respiration Rates*

Soil respiration has a strong spatial dependence ($C_0/(C + C_0) = 0.11$) [52], and stable environmental factors such as soil physicochemical properties and vegetation characteristics play a leading role in the spatial heterogeneity of Rs. The CV of Rs (86.21%) calculated in this study is larger than those of previous studies. For example, the CV of the Rs of poplar plantations at different growth stages along the Yili River ranged from 5.7% to 42.6% [53]. Yan et al. observed Rs and various factors in larch plantations in Pangquangou Nature Reserve using the hierarchical grid nested method based on three research scales, which found that the CV of Rs was 30.27–36.11% [54]. One study based in Taiyuan basin found that the CV of Rs was 42–59%, which showed obvious seasonal variation as well as patchy and continuous distribution [55]. In this study, the discontinuous patch distribution of Rs may be closely related to the uneven distribution of water and salt content (Figure 2), soil water content showed patchy and blocky distribution in the Ebinur Lake Basin ($\alpha = 15.50$, CV > 10%), and the soil surface water content was positively correlated with salt content [38].

In this study, the Rs ($0.29 \pm 0.25$ μmol·m$^{-2}$·s$^{-1}$) calculated was relatively small compared with the results obtained for regions with abundant precipitation. For example, the average Rs values of Lingkong Mountain Nature Reserve in Shanxi Province were 2.751 μmol·m$^{-2}$·s$^{-1}$, with a fluctuation range of 1.19–4.41 μmol·m$^{-2}$·s$^{-1}$ [15]. The fluctuation range of Rs in the Mao Village of Guilin was 1.39–5.31 μmol·m$^{-2}$·s$^{-1}$ [24]. The average value's distribution ranges of Rs in desert ecosystems, desert–farmland transition ecosystems, and farmland ecosystems in Gurbantunggut Desert were $0.33 \pm 0.06$ μmol·m$^{-2}$·s$^{-1}$, $0.33 \pm 0.16$ μmol·m$^{-2}$·s$^{-1}$, and $0.147 \pm 0.008$ μmol·m$^{-2}$·s$^{-1}$, respectively [56]. Additionally, the Rs in the desert–oasis transition zone near Minqin Desertification Station had mean values of 0.53, 0.24, and 0.18 μmol·m$^{-2}$·s$^{-1}$ in May, August, and November, respectively [16]; this result is similar to those of this study, and the reason may be that the habitat in Minqin Desertification Station was analogous to the study area and the vegetation composition and climatic conditions of the areas were similar, so the Rs values were relatively similar.

*4.2. Relationship between Soil Respiration and PC1*

There was significant correlation between Rs and PC1 ($R^2 = 0.89$, $p < 0.01$). PC1 combines the information from the morphological traits of plants and soil water content, which was significantly positively correlated with maximum height, leaf width, leaf dry matter content, leaf length, leaf thickness, specific leaf area, and soil water content ($R^2 > 0.56$, $p < 0.01$). Among them, the angle between maximum height, leaf width, and Rs were small and the direction was similar; the leaf dry matter content and Rs were almost inversely parallel, the length of the leaf dry matter content was long; and the length of the soil water content was very long. These variables were all related to Rs (Figure 4).

Rs was significantly positively correlated with maximum height ($p < 0.01$). Previous studies have shown that maximum height is related to the strategy by which plants obtain organic matter [43]. Plants increase maximum height to increase light-use efficiency to obtain greater resources and biomass [57]. In this study area, drought conditions and water–salt stress were the limiting factors affecting the maximum height.

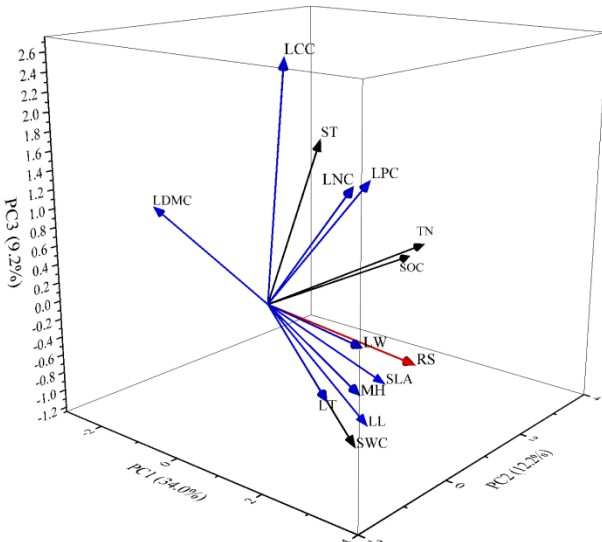

**Figure 4.** MFA ordination of soil respiration and community functional traits. (Legend: The direction of the arrow indicates the positive and negative correlations between the research variables and the sorting axis, and the length of the arrow line represents the strength of the relationship between research variables and sorting axis. Rs−soil respiration, MH−Maximum height, LL−Leaf length, LW−Leaf width, LT−Leaf thickness, LDMC−Leaf dry matter content, SLA−Specific leaf area, LCC−Leaf carbon content, LNC−Leaf nitrogen content, LPC−Leaf phosphorus content, ST−Soil temperature, SWC−Soil water content, TN−Total nitrogen, and SOC−Soil organic content).

Rs was significantly positively correlated with leaf width ($p < 0.01$). Leaf width, leaf length, leaf thickness, and specific leaf area had significant effects on plant growth rate. Leaf area was significantly positively correlated with direct and diffuse radiation fluxes [58]. In order to adapt to the drought environment, plants absorb nutrients from the surrounding environment constantly, thus increasing the specific leaf area; however, plants have a large demand for light in this process, and the demand for nutrients and water stored in leaves is small. Therefore, the utilization efficiency of nutrient resources is relatively low, the leaves of plants are thinner, and the leaf tissue density is smaller [59]. The larger the specific leaf area, the larger the capture area, resulting in a stronger ability of the plants to obtain resources as well as to make abundant organic substances required by animals and microorganisms. Such a large area also promotes an increase in litter input and enhances the activity and metabolism of microorganisms [60]; thus, Rs increases with specific leaf area. Plants have complex physiological responses in adversity, as observed in this study. The leaves of rare-salt plants *Salsola collina* and *Suaeda microphylla* were highly fleshy. *H. ammodendron* evolved an assimilation clade. The leaves of *Lycium ruthenicum Murr*, *N. schoberi*, *R. soongorica*, and others were highly degenerated. Additionally, all had small specific leaf area and reduced transpiration, which prevented water loss in a high-intensity light environment, which is beneficial for plants to maintain water and nutrients to increase drought resistance while reducing the ability to obtain light and carbon resources, and Rs decreased synchronously. In recent decades, the water volume of the Ebinur Lake has decreased significantly, desertification is more severe, the ecological environment is poor, small specific leaf area can reduce the contact area with the environment so that plants are protected from damage, and Rs is correspondingly limited.

Rs was significantly negatively correlated with leaf dry matter content ($p < 0.01$). In this study, the degree of variation in leaf dry matter content was weak, indicating that plants tended to choose similar evolutionary methods to adapt to the environment of water stress and high transpiration. Leaf dry matter content reflects the adaptability of plants to stress habitats [61] and can predict the ability of plants to obtain environmental resources stably [62,63]. When plants are in drought-stress habitats, they invest more resources in leaf construction to increase the distance and resistance of water diffusion from the interior

of the leaf to the leaf surface [64–67] to change the gas conductivity of the blade. In desert ecosystems, water is a key factor in the decomposition and transformation of soil organic matter [68]. Drought stress has a large limitation, and a larger leaf average tissue density corresponds to a larger leaf dry matter content to slow down the growth and metabolism rates and to improve the ability to resist physical damage.

In addition to the abovementioned key plant function traits, Rs was significantly positively correlated with soil water content. The heterogeneity of soil water content was strong in this study ($\alpha = 15.50$, CV > 10%), and Rs was similar to soil water content in the spatial distribution (Figure 2). Han et al. found that the increase in Rs (19–75%) by rainfall was much larger than that by warming (11.4%), indicating that soil water content could explain about 90% of the change in Rs in sandy land [24]. The effect of soil water content has been described by several equations, including linear, logarithmic, quadratic, and parabolic functions of soil water expressed as matric potential, gravimetric water content, volumetric water content, fractions of water holding capacity, water-filled pore space, precipitation indices, and depth to water table [69].

### 4.3. Relationship between Soil Respiration and PC2

PC2 was significantly positively correlated with total nitrogen and soil organic content ($R^2 > 0.9$, $p < 0.01$), but no significant correlation was found between Rs and PC2 ($p > 0.05$). The content of soil organic content and total nitrogen affects the conversion of soil organic matter and the formation of some nitrogen elements. Based on existing research results, soil organic content is the carbon source of soil microbial respiration, and nitrogen promotes plant growth and provides more substrates for Rs [70]. Soil organic content is the main factor affecting the spatial distribution of Rs [71]. Sufficient nutrients in the soil are also important for the decomposition and respiration of microorganisms and animals. In this study, Rs was significantly positively correlated with total nitrogen and positively but not significantly ($p > 0.05$) correlated with soil organic content. The reason for this may be that the heterogeneity of soil organic content was weak ($\alpha = 30.0$, CV > 10%), This is consistent with the spatial distribution map (Figure 2).

### 4.4. Relationship between Soil Respiration and PC3

There was significant correlation between Rs and PC3 ($R^2 = 0.53$, $p < 0.01$). PC3 combines the information from physiological traits of plants and soil temperature, which was significantly positively correlated with leaf carbon content, leaf nitrogen content, leaf phosphorus content, and soil temperature ($R^2 > 0.51$, $p < 0.01$).

Rs was significantly positively correlated with leaf carbon content, leaf nitrogen content, and leaf phosphorus content ($p < 0.01$). In this study, their range of variation was weak, and the stable state may be that plants have a strong ability to adapt to poor soil conditions and to maintain relatively stable nutrients, which are ecological strategies to adapt to poor environments. The adaptation of leaf carbon content, leaf nitrogen content, and leaf phosphorus content to heterogeneous environments are common results of phenotypic and genetic differentiation and forms different physiological ecology and nutrient utilization characteristics during long-term evolution [59]. The leaf eco-chemical characteristics of desert halophytes in the Ebinur Lake are affected by soil salinity and element content [72]. According to an ecological stoichiometry analysis, the correlation coefficient between leaf carbon content and Rs is the smallest, which is related to the carbon accumulation by stomatal conductance and dark reaction at night under daytime-high temperatures. In the Ebinur Lake desert ecosystem, the leaf nitrogen contents of both *H. halodendron* and *N. schoberi* were significantly lower than the average leaf nitrogen content of grasslands in China [73], and lower leaf nitrogen contents restricted Rs in the study area; plant growth was limited by both nitrogen and phosphorus [62], but soil salinization had little effect on the absorption capacity of plants [35]. Therefore, when competition pressure is low, plant leaves increase investment in nitrogen to meet the higher demand for nitrogen [74].

Rs was positively but not significantly ($p > 0.05$) correlated with soil temperature. Rs and heterotrophic respiration were significantly correlated with soil temperature at depths of 2.5 and 10 cm ($p < 0.001$) [13], and the relationship between soil temperature and Rs in different research sites may be different, but the influence of temperature on Rs usually can be expressed by an exponential model [69]. However, the heterogeneity of soil temperature was weak in this study ($\alpha = 25.50$, CV < 10%), and the temperature effect manifested only when there was sufficient soil moisture to permit significant root and microbial $CO_2$ production [75]. Under extreme drought conditions, the average annual rainfall of Ebinur Lake Basin is only 105 mm, which further proves the above inference.

The above analysis shows that plant functional traits achieved synergy and trade-off between each other by regulating resource allocation in the long-term adaptation process [76].

## 5. Conclusions

Our findings suggest that soil respiration, as one of the ecosystem functions, expound on its interaction and have an effect on plant functional traits, which can predict the change in pattern in the arid desert area. This study points out that environmental factors and biological factors jointly drive the spatial dynamic change of soil respiration, which is helpful for understanding the problem of "missing carbon sink" in the current carbon balance. Our research also can increase the limited knowledge of the subject at present.

This study evaluated soil respiration under the influence of easily measurable plant community characteristics and environmental factors in the Ebinur Lake Basin due to the influence of global changes and human activities. The uncertainty is more obvious, and the influencing factors are more complex; therefore, further studies will be undertaken on the main factors affecting Rs in the ecosystem.

**Author Contributions:** Conceptualization, G.L.; methodology, G.L.; software, J.W. and H.S.; validation, H.S.; formal analysis, J.W., H.S. and X.H.; investigation, J.W. and X.H.; resources, H.S.; data curation, J.W. and X.H.; writing—original draft preparation, H.S.; writing—review and editing, J.W.; visualization, H.S.; supervision, G.L.; project administration, G.L.; funding acquisition, G.L., X.H. All authors have read and agreed to the published version of the manuscript.

**Funding:** This research was jointly funded by the. National Natural Science Foundation of China: (No. 31560131 and 31760168), the Xinjiang Uygur Autonomous Region university scientific research project (No. XJEDU2020I002), and the Xinjiang Uygur Autonomous Region Graduate Research and Innovation Project (No. XJ2019G020).

**Institutional Review Board Statement:** Not applicable.

**Informed Consent Statement:** Not applicable.

**Data Availability Statement:** All relevant data for this study are reported in this article.

**Acknowledgments:** We would like to thank Lamei Jiang, Zhoukang Li, Wenjing Li, Fang Yang, Deyan Wu, Yudong Chen, Feiyi Liu, Zhiqiang Li, Yulin Shu, Suwan Ji (College of Ecology and Environment, Xinjiang University) for their help in field work. And we are grateful to (www.letpub.com (accessed on 25 March 2022)) for their linguistic assistance during the preparation of this manuscript.

**Conflicts of Interest:** The authors declare no conflict of interest.

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
