# Peer review of "The Relationship between Soil Respiration and Plant Community Functional Traits in Ebinur Lake Basin"

_agronomy, doi:10.3390/agronomy12040966_

Round 1

Reviewer 1 Report

The Authors measured soil respiration once in a 1 ha plot in an arid desert region and studied its relationship with plant functional traits. The measurement is spatially very detailed and this topic is important to reveal the connection between soil processes and plant traits. However, I think that the measurements were not enough to adequately study the question and the manuscript is insufficient for a scientific paper in this form.

Generally, I think that the Authors conducted too few measurements, I mean they should have measured also other variables (such as soil organic carbon, total nitrogen or soil moisture) and the measurements should have been repeated, as the Authors also pointed out that seasonality is an important aspect. In addition, the statistical analyses were redundant and contained groundless results. Finally, the justifications and explanations in the Discussion section were scientifically weak.

My suggestion is that the Authors should repeat all the measurements in the same place in another season, and measure new parameters as well. In addition, I suggest to reconsider the statistical analyses. I hope that my detailed comments will help to develop the manuscript.

Detailed comments

Title

line 2: Community Functional

Abstract

Some more background information is needed. E.g. about the relationship between soil respiration and plant traits.

The Authors should also point out their goals.

lines 16-18. These “weak” and “medium” variation degree is unknown. What does it mean? What is weak, what is medium? Where is the scale from?

I also missed the discussion part from the Abstract. At least one sentence is needed to describe the point of the results.

Introduction

Generally, I really missed the literature. There are many statements which had no literature supports. For instance: lines 30, 47-51, 52-60, and 64-70. The Authors stated that no study has been conducted about the topic. Let me suggest some literatures for reading: Metcalfe et al 2011, Biogeosciences(8) 2047-2061, Buchanan et al 2021 Journal of Environmental Quality(51)33-43; Johnson et al 2008, Biology Letters(4) 345-348.  And so on. The Authors should study the literature more deeply.

line 29: Soil respiration changes the biogeochemical cycle and ecosystem stability? How? When? Why?

line 36: Soil organic carbon (and “whole” soil water content) affect the development of soil organic carbon? Does it make any sense?

line 38. If soil organic carbon and total nitrogen are so important in this topic, why didn’t the Authors took them into consideration in their study?

line 39 The same is for soil water content as for organic carbon and total nitrogen. I really miss the data about these variables in this study. However, the Authors know that these variables are important.

line 44: Plants adjust?

line 59-60 Reconsider this statement.

lines 69-71: I do not think that this paragraph justice the goals of the study. My problem is that the Authors didn’t connect soil respiration, plant traits and extreme landscape together. Briefly, why is it interesting to measure soil respiration and plant traits in a desert?

lines 75-76. These methodological details are unnecessary here.

Materials and Methods

lines 92-95 (or Table2): species names need their descriptor and the description year.

lines 112-113: were the two pieces of equipment calibrated together? Wasn’t any chance to have bias due to the two LI-8100s.

lines 123-125 What are “multiple degrees”?

line 126: Are five pieces of leaves correctly representative for a 25 m2 plot? Isn’t it needed to choose amount of leaves proportional to the species’ dominance? Don’t “healthy” leaves make some bias about the real trait values? (see line 289)

Table 1: I miss other literature. Only LDMC, SLA and LPC were supported by references.

lines 147-148: By community weighted mean, we take into account all the community, not only the dominant species. What does “each dominant species” mean here? It should be described more detailed.

lines 150. How did the Authors measure the biomass weight of each species?

line 153-155: I do not think that CV is a relevant and interesting statistic.

Statistical methods:

The analyses are redundant, not justified and superficially described. And no diagnostic tools were used.

line 168 Did the Authors really need statistics to find out the species number?

line 170 What are plants’ specific categories? This information is not mentioned any more.

line 174 and line 183 and Table 3: contradiction

line 175: What does ±0.25 mean? Standard deviation? CV? cf. Table 3.

line 175, 178, 179 When is variation moderate? When is weak? Where did the Authors get this scale?

Table 3: Does it have any sense to display all these data? What is “crest value”?

line 186 I suggest to look for a more exact statistical method which could reveal the “similarities” between the spatiality of variables. This is insufficient to display some plots and saying “characteristics of patch distribution were generally lower in the west …” It is not enough for a scientific paper and make not so much sense. What does it exactly mean? In this form, these sentences are subjective not objective.

MFA

Multiple factor analyses are used to reveal the relations of two groups of variables. In this study, the Authors have one variable of soil respiration and a group of plant trait variables. In this form, it would be better to use PCA.

It is redundant to use besides MFA, linear regression and then a stepwise regression analysis. It is clear, that many plant traits are correlated with each other. It makes no sense to take all the correlated variables into one regression model. And what about model diagnostics?

Please, reconsider all the statistical analyses.

Discussion

lines 240-243: This paragraph should be eliminated.

lines 245-257: It was not justified, why the CV value is important. What does it indicate? However, when seasonality is so important (line 251), why did the Authors measure only once in a year? Can we draw any conclusions from only one measure occasion? If yes, please, justify it!

line 262-264: “mean distribution ranges” means or ranges? If these data are ranges, then the ± symbol is incomprehensible. If they are mean values with SD or CV, then these data run into negative range?

lines 258-272: I do not think that this part properly interprets the results.

line 274: it is very confusing to see everywhere these abbreviations (LW, LL, MH etc). At least, in the Discussion section, use the whole name of traits.

lines 275-276: how do the tree trunks come to here? Did the Authors investigated the distance between tree trunks and soil respiration? My main problem with the Discussion section is that the results and the reasoning are not always related.

line 277 pH?

line 281: Did the Authors measure plant growth rate?

line 289: these plants had “highly degenerated” leaves. Then how could the Authors sample “healthy leaves?

line 308-309. It is interesting that the Authors found that there was significant correlation between soil respiration and LCC, as we can see in figure 3 that these two variables are almost perpendicular…

There are also other problems, but I gave up to consider all of them. This study also needs a significant English grammar correction.

Reviewer 2 Report

The article is well designed nicely written. The following points need to be addressed.

  1. The abstract must be strengthened with the inclusion of results in qualitative form.
  2. The introduction should be shortened and reorganized in view of the identified problem.
  3. Methodology is ok
  4. Results well presented
  5. Discussion must be supported with newly published reports
  6. The conclusion must be reformed with a clear message and recommendation of the study.
  7. The language of the article is fairly ok but critical rechecking is suggested.
  8. Fram the references as per the style and pattern of the journal. cross-checking suggested.

Round 2

Reviewer 1 Report

General

Previously, I suggested the Editor to reject this manuscript because it had too many trivial and serious mistakes and it had not enough data for a scientific paper. The Authors found more related data (somehow? somewhere?) and that is why I reviewed again this MS. The manuscript was seriously promoted, however, I will suggest major revision. The amount of data is now enough and the applied methods are now better. The problems are now the wording and the understanding of the new methods and results. My detailed suggestions can be found down.

Detailed comments (line numbers refer to the document without track changes)

General:

Next time, it would be nice from the Authors to refer all the changes/corrections with line numbers. I found some line numbers but not everywhere.

I am really interested in why the Authors did not add the environmental variables to the previous version of the study.

Abstract

(Previous answers are accepted)

lines 10-12:  This introduction was very good. It would be an interesting connection between the study and climate change and carbon balance. However, the Authors never mentioned this topic again and it was strange and an unused potential.

lines 17-19: I did not like these abbreviations. There are too much and it is difficult to remember all of them. Do not forget that the readers will first meet these names and (s)he won’t be so patient to learn all of them. It is not a problem to use the names of these variables. And it is obligatory in the Abstract to write them wholly because the Authors use them here for the first time.  If the Authors think that the list of the names would be too long, mention only 3 or 4 of them, the most important ones. It is also enough.

lines 21-22. Affecting is a strong phrase. The Authors found correlation, but it does not prove the effect. Please, find a weaker term: such as connection, relationship etc.

line 22: Maximum

Introduction

(Previous answers are accepted)

line 37: What kind of trophic interactions? We are talking about plants, aren’t we?

lines 49-53: too long sentence! lines 47-50 should be one sentence.

lines 53-64: Literature reference needed.

lines 66-73 I would connect this paragraph with soil respiration.

lines 77-80: Add a predicate to the sentence!

Material and methods

(Previous answers are partly accepted)

Point 18

This reference was not an answer to my questions. Can the Authors mention another study which uses this method?
(However, the referred paper cannot be found on the internet. Only a citation was found in Google Scholar, but with other authors…)

Point 20

It was not in line 162, and it was not „each species” but rather „of all species”…

Point 21

It was not removed! See line: 161 however, the Authors admitted that it was a mistake.

New questions for this Section:

It is not clear, in which units do the Authors calculate the Tc and CV values. They are writing about plots, but there are 5×5 m plots, and three 1×1 m plots in „each plot”. It is not clear how these plots are standardized for calculating Tc and CV for trees and herbaceous plants.

line 178: N(h) not n(h). What is h? What are research variables? CWM? or CV or what? Numerical logarithm?

line 186: It should be explained, what is 0.25 or 75%... I can see in Table 3,  that it must be the proportion of C/(C+C0) but the Authors should describe the method clearer. Some words could be added about the Variogram types as well.

line 194: What does it mean that the variables were simulated by a softwer?

line 196: what kind of "different models" were compared to obtain the optimal semi-variance function model of each index (what is an index?)

lines 198-200 Please, reconsider this sentence.

line 200 Why did the Authors forwardly select three axes in the PCA?  It is a result. We cannot say forwardly that we will have three important axes.

Results

(Previous answers are partly accepted)

Point 24

I just wanted to say that we do not need any statistics to write down the numbers of different taxa. Statistics are e.g. mean value, standard deviation etc. but not the sum of something.

Point 27

The Authors wrote me in their letter that they corrected their mistake but it is not true… line: 213.

New questions for this Section:

line 214-215: what do the Authors mean that the values were different. Of course, they were different, they were measured in different units.

line 218-221: What is a decision coefficient and where are these values?

line 222: better than what?

lines 225-227: The Authors write about nugget values but in Table 3, there are only “Proportion”  values. I have a sort of idea that the Authors do not really know and understand how the methods they use work and what are these numbers they got. At least, they do not know the meaning of the different terms.

line 229-234: these sentences should be taken into the Methods section.

Table 3. The table contains the model type but the Authors did not mention this information. Why is it so? If it is not interesting, why is it in the table?

lines 247-248: Reconsider this sentence.

line 250: What is a spatial distribution relationship? Is it measured with similar range values?

line 251: What do the Authors mean about polar opposite?

line 251: Rs or Rs rate?

line 251: How could a value (“Rs rate”) be similar to spatial distribution patterns?

line 253: What kind of significant differences? How did the Authors calculate it?

lines 263-264: 34%, 12.2% and 9.2% are not correlation coefficients. I’m afraid that the Authors do not understand what the results show them.

line 266: What does it mean that the “indicators” (what are indicators?) can better reflect the relationship between Rs and […]? Better than what?

Rs or RS?

line 270: “represent” is not the proper term in this context. And what about SLA?

lines 270-284: Reconsider this part. What are “increasing trends”? Please, read some similar analyses in other papers and look up the proper terms and phrases for this type of analysis.

line 279-280: This wording is confusing. The PCA equally treats the variables, it does not analyse the relationship between Rs and the other variables, it analyses among all the variables.

line 280: This sentence is not true. What about LCC? What about TN and SOC?

Table 4: The 5-7th columns are redundant.

Figure 3: The meaning of the abbreviations is missing. The meaning of the colours is missing.

line 289-291: I do not think that these equations are necessary.

line 292: Why did not the Authors mention in Methods that they want to use Pearson correlation?

line 292-301: Please, avoid the word “very significantly”.

Discussion

(Previous answers are accepted)

line 315: What do The Authors mean “stable”? Stable in time? Or how?

line 318: I do not think that C/(C+C0)=0.89 is the explanation of the statement of this sentence.

line 319-321: Please, read this sentence one more time.

line 327: different from what?

line 328: However, the water content is not as various as the Rs…

line 337: this “dessert ecosystem” occurred also in the previous MS version.

I hardly believe that the Authors read this part again… I suggest to read again the text and reconsider also the useless capitals in the middle of sentences and the using of commas instead of full stops.

line 346: please, avoid “very significant(ly)”. Something is statistically significant or not.

line 351 Please, use height, not MH. Or plant height.

line 351: Which studies?

line 353: Literature?

line 355: there are too much abbreviations. Please, use the real names of the variables.

Conclusions:

Statements 1-4 are not conclusions. They are the repetition of the results. I suggest the Authors to leave the Conclusion part or think again about the real conclusions.
